# Influence of Surface Cleaning on Quantum Efficiency, Lifetime and Surface Morphology of p-GaN:Cs Photocathodes

**DOI:** 10.3390/mi13060849

**Published:** 2022-05-29

**Authors:** Jana Schaber, Rong Xiang, Jochen Teichert, André Arnold, Petr Murcek, Paul Zwartek, Anton Ryzhov, Shuai Ma, Stefan Gatzmaga, Peter Michel, Nikolai Gaponik

**Affiliations:** 1SRF-Gun Group, ELBE Department, Institute of Radiation Physics, Helmholtz-Zentrum Dresden-Rossendorf, 01328 Dresden, Germany; r.xiang@hzdr.de (R.X.); j.teichert@hzdr.de (J.T.); a.arnold@hzdr.de (A.A.); p.murcek@hzdr.de (P.M.); p.zwartek@hzdr.de (P.Z.); a.ryzhov@hzdr.de (A.R.); s.ma@hzdr.de (S.M.); s.gatzmaga@hzdr.de (S.G.); p.michel@hzdr.de (P.M.); 2Eychmüller Group, Institute of Physical Chemistry, Technische Universität Dresden, 01062 Dresden, Germany; nikolai.gaponik@tu-dresden.de

**Keywords:** p-GaN, UV-photocathode, quantum efficiency, surface cleaning, surface morphology

## Abstract

Accelerator scientists have high demands on photocathodes possessing high quantum efficiency (QE) and long operational lifetime. p-GaN, as a new photocathode type, has recently gained more and more interest because of its ability to form a negative electron affinity (NEA) surface. Being activated with a thin layer of cesium, p-GaN:Cs photocathodes promise higher QE and better stability than the known photocathodes. In our study, p-GaN samples grown on sapphire or silicon were wet chemically cleaned and transferred into an ultra-high vacuum (UHV) chamber, where they underwent a subsequent thermal cleaning. The cleaned p-GaN samples were activated with cesium to obtain p-GaN:Cs photocathodes, and their performance was monitored with respect to their quality, especially their QE and storage lifetime. The surface topography and morphology were examined by atomic force microscopy (AFM) and scanning electron microscopy (SEM) in combination with energy dispersive X-ray (EDX) spectroscopy. We have shown that p-GaN could be efficiently reactivated with cesium several times. This paper systematically compares the influence of wet chemical cleaning as well as thermal cleaning at various temperatures on the QE, storage lifetime and surface morphology of p-GaN. As expected, the cleaning strongly influences the cathodes’ quality. We show that high QE and long storage lifetime are achievable at lower cleaning temperatures in our UHV chamber.

## 1. Introduction

Free-electron laser (FEL) and terahertz (THz) radiation sources are powerful tools with a wide application in biomedicine, security imaging and condensed matter physics [1,2]. Both tools require high brightness electron beams, which are generated by so-called photocathodes. Photocathodes should provide a high quantum efficiency (QE), a low thermal emittance, a fast response and a long operational lifetime. None of the state-of-the-art photocathodes can meet all the desired requirements to be a “perfect” photocathode so far. Therefore, new materials that can be used as potential photocathodes are urgently needed for future developments in accelerator research.

Pure metals, such as magnesium and copper, are widely used as photocathodes. Their preparation is easy and they work robustly, but they do not provide high quantum efficiency (QE) [3,4]. Contrarily, semiconductor photocathodes, such as cesium telluride (Cs_2_Te) or potassium cesium antimonide (K_2_CsSb), offer more advantages than metal photocathodes [5,6]. Both mentioned semiconductor photocathode types could provide a QE of 5–20%, a long lifetime and a low thermal emittance. However, these photocathodes are very sensitive to any vacuum instability, and their preparation is complex. Although gallium arsenide (GaAs) photocathodes have many disadvantages, such as being difficult to prepare, a low robustness and a short lifetime of only several hundred hours, they are the only photocathodes providing spin-polarized electrons [7,8,9].

Thus, searching for new materials with better photoemitting properties led to the promising field of gallium nitride (GaN) photocathodes [10,11].

III-V semiconductor materials with a wide bandgap, especially GaN, are famous for their application in ultraviolet (UV)-sensitive detectors and in light-emitting diodes (LEDs) [12]. With the activation of a low work function alkali metal, such as cesium, III-V semiconductors are able to reduce the surface work function to form a negative electron affinity (NEA) surface [13]. P-doping (e.g., magnesium as a dopant atom) of the semiconductor is required to introduce band-bending so that photoelectrons can overcome the energy barrier more easily [14,15].

The p-GaN:Cs photocathode promises to be more inert against poisonous gases such as oxygen and is able to reach QE values of 50–70% [16,17]. The activation of p-GaN is similar to the activation of GaAs photocathodes, where a so-called “yoyo” process is applied. During this yoyo process, oxygen (O_2_) and cesium (Cs) are alternatively applied several times until a final photocurrent maximum is reached [18,19,20]. However, the absence of oxygen in the preparation of p-GaN:Cs photocathodes has been discussed and demonstrated in simulations and experiments [15,21]. Therefore, a successful activation for p-GaN photocathodes exclusively with cesium is possible and presented in this work. Achieving a QE of about 10% by exclusively using cesium with a high reproducibility is a good basis for future p-GaN:Cs photocathode research. One of the most important steps for achieving the required NEA surface and thus a high QE is to guarantee an atomically clean surface before cesium is deposited [22,23,24].

In this paper, we show the influence of different cleaning steps on the p-GaN surface and the photoemission performance (QE and storage lifetime) of p-GaN:Cs on two substrate materials, sapphire and silicon. The aim of this study is to identify the most efficient cleaning method leading to a high QE and a long lifetime.

## 2. Materials and Methods

The commercially available p-type GaN samples were grown by using metal–organic chemical vapor deposition (MOCVD) on sapphire (Al_2_O_3_) or silicon (Si) with an average magnesium concentration of 6 × 10^16^ to 1 × 10^17^ cm^−^³ [25].

For p-GaN on sapphire, no buffer layer in between is necessary, while for p-GaN on silicon, an aluminum nitride (AlN) buffer layer was used to reduce the lattice mismatch. The received 2-inch wafers were diced into 9 × 9 mm² squares to fit our experimental sample holder in the UHV chamber.

Our cleaning process consists of several steps, starting with a wet chemical cleaning, followed by a vacuum thermal cleaning at various temperatures, going to the activation with cesium and ending with the measurement of the storage lifetime.

Before and after the wet chemical cleaning, atomic force microscopy (AFM) was applied to investigate the samples. The AFM images were taken on Asylum Research Cypher AFM Microscope in AC mode (tapping mode) with an AC-160 cantilever (silicon tip with a diameter of 10 nm and silver coating).

The wet chemically cleaned p-GaN samples were transferred in a glovebox with a dry nitrogen atmosphere. The samples were mounted with steel washers on our sample holder and were transported under a nitrogen atmosphere into an ultra-high vacuum (UHV) chamber, as shown in Figure 1. The important requirement for an efficient cleaning is to avoid any oxygen contamination before, during and after the activation with cesium. In the UHV chamber, a thermal cleaning was carried out using a 400 W halogen lamp with a metal reflector. The temperature on the p-GaN surface was measured with an infrared (IR) sensor (OPTRIS, CT 3MH1-CF3-CB3, Berlin, Germany), which was calibrated by the supplier.

Evaporators from SAES (Milan, Italy) were used for the cesium deposition on the cleaned p-GaN surface. The dispenser released a constant cesium flux, causing a pressure of 1 × 10^−9^ mbar, by applying a current of 3.3–3.5 A through it.

During the activation exclusively with cesium at room temperature, the p-GaN sample was illuminated by ultraviolet (UV) light at 310 nm wavelength and 110 µW power (ROITHNER, DUV310-HL5N, Wien, Austria). An aperture in front of the UV-LED was used to reduce the incident light power. The extracted photoelectrons from the p-GaN:Cs photocathode were collected by a copper ring anode with a positive bias of 150 V. The photocurrent during the cesium deposition was recorded in situ, and the deposition was stopped when the photocurrent reached a maximum value.

The QE was calculated by Equation (1), where *h* is the Planck constant, *c* is the speed of light, *q*_e_ is the elementary charge of an electron, *λ* is the wavelength, *P*_Light_ is the power of the drive light on the cathode and *I* is the photocurrent of the p-GaN:Cs photocathode [26].
(1)$$\mathrm{QE}=\frac{h\cdot c}{q_{e}}\cdot \;\frac{I}{\lambda \cdot P_{\mathrm{Light}}}\;$$

After several thermal cleanings and activations, the surface morphology of the p-GaN:Cs photocathode was examined by scanning electron microscopy (SEM) combined with energy-dispersive X-ray spectrometry (EDX). The SEM images were carried out on a Zeiss NVision 40 FIB/SEM microscope (Carl Zeiss GmbH, Jena, Germany) with an electron beam energy of 10–20 kV and a secondary electron (SE) detector. The EDX measurements were performed with a Bruker QUANTAX EDS spectrometer (Billerica, MA, USA).

## 3. Results

### 3.1. p-GaN on Sapphire

#### 3.1.1. Surface Study of Native Samples

The surface topography before and after the wet chemical cleaning was studied by atomic force microscopy (AFM). The crystal growth and possible dislocations in the crystal lattice of semiconductors were detectable with AFM investigations.

Figure 2 shows the surface of an untreated as-received p-GaN on sapphire sample. The p-GaN sample exhibits a smooth, well-defined, uniform surface. The average root mean square (RMS) roughness is 1.7 nm for the examined area in Figure 2a. Bright spots that correspond to intersections of screw-component dislocations can be found all over the surface. These dislocations create step terminations and cause lattice defects, which are the reason for hillock formations [27].

Additionally, dark spots were detected on the p-GaN surface. These correspond to dislocations in the crystal growth, as shown in Figure 2b. This so-called etching pit exactly represents the hexagonal crystal structure of the GaN lattice and becomes visible when the surface is etched with oxidizing detergents [28,29,30]. The observed pit has a width of 360 nm and a depth of several micrometers, which indicates that these hexagonal void holes result from a poor coalescence at the beginning of the crystal growth.

#### 3.1.2. Wet Chemical Cleaning

The p-GaN samples were processed by a wet chemical cleaning in order to guarantee a successful activation. The purpose of the wet chemical cleaning is to remove residual attached contaminants, such as air molecules or dust particles, from the surface.

The p-GaN samples were cleaned in 99% pure ethanol in an ultrasonic bath for 15 min. Afterward, the samples were treated with a mixture of sulfuric acid (98%) and hydrogen peroxide (30%) (1:1 by volume), a so-called “piranha” solution, at 140 °C for 15 min. In the last step, the samples were dipped into hydrochloric acid (36%) and rinsed with deionized water and 99% ethanol. The piranha solution is a strong oxidizing agent and has the ability to remove a majority of organic contaminants, whereas the hydrochloric acid (HCl) removes metal ions from the surface. The effect of the wet chemical etching process was studied by AFM measurements.

The wet chemically cleaned p-GaN surface is illustrated in a 2D and a corresponding 3D AFM image in Figure 3a,b, respectively. The imaging was started immediately after the cleaning and was completed within 35 min from top to the bottom under normal atmospheric conditions.

At the beginning (on the top of the AFM image), the surface was still smooth and had many usual hillocks, representing screw dislocations in the hexagonal crystal structure. At this stage, the surface had an average RMS roughness of 1.7 nm, similar to the untreated sample in Figure 2a. However, during the ongoing measurement, some additional bright peaks became visible, whose appearance was even more pronounced with the measurement time. At the end of the measurement (bottom in Figure 3a), the surface was completely covered with these bright peaks with a maximum height of about 77 nm, illustrated more clearly in the 3D AFM image (Figure 3b). We assume that the wet chemical cleaning uncovers free, unbound gallium atoms on the p-GaN surface. With the exposure to atmospheric oxygen, these free gallium atoms start to form gallium oxide (Ga_x_O_y_) islands. With increasing time, these islands are converted into a homogeneous oxide layer that covers the complete p-GaN surface. Therefore, the importance of oxygen exclusion can be seen in these experiments.

In the following experiments, all samples were immediately prepared and transported under a nitrogen environment after the wet chemical cleaning to prevent this oxidization process.

#### 3.1.3. Influence of Thermal Cleaning

After the wet chemical cleaning, the samples were thermally cleaned at different temperatures to generate an atomically clean p-GaN surface. Afterward, the samples were activated with cesium to obtain an NEA surface and thus produce p-GaN:Cs photocathodes.

Figure 4 shows a comparison of the gained QE values of three p-GaN:Cs on sapphire photocathodes, depending on the cleaning temperature. The cycles of heating and activation were repeated several times for each sample in order to improve the statistical accuracy.

The ordinate in Figure 4 shows the QE of the fresh activated p-GaN:Cs photocathodes, whereas the x-axis shows the sample number, followed by its cycle number. The color represents different temperatures used in the thermal cleaning. We do not show samples A1–A3 here; they were thermally treated at over 600 °C, leading to results similar to those found for sample A4.

The highest QE (>10%) was achieved for sample A5 (red) when the p-GaN was cleaned at 500 °C. Thermally cleaned at a temperature above 600 °C, sample A4 (violet) reached lower QE values than those obtained at a lower thermal cleaning temperature (400–500 °C). In order to determine the reason for the low QE after the thermal cleaning at high temperature, surface studies with SEM were carried out (Section 3.1.5 below).

A comparison of the measured storage lifetimes of the p-GaN:Cs photocathodes is shown in Figure 5. We define the storage lifetime as the time interval from the completion of the activation to the time when the QE drops down to 1%. For reasonable comparison, the QE was tracked for several days, and a prediction of the lifetime has been made from its exponential decay fitted by a suitable tri-exponential model, shown in Equation (2) [31].
(2)$$y=y_{0}+A_{1}\cdot e^{-\frac{x-x_{0}}{t_{1}}}+A_{2}\cdot e^{-\frac{x-x_{0}}{t_{2}}}+\;A_{3}\cdot e^{-\frac{x-x_{0}}{t_{3}}}$$

The activated p-GaN:Cs photocathodes were stored under a stable UHV environment of 1 × 10^−10^ to 5 × 10^−10^ mbar during the lifetime measurement. The fluctuations in this vacuum range are considered normal and without effect on the p-GaN:Cs photocathodes’ quality.

The p-GaN:Cs on sapphire sample A4, cleaned above 600 °C, shows storage lifetimes of less than 50 h. This means that after the cesium deposition, the QE decayed rapidly to 1% within 50 h. This photocathode survived longer after the first activation than after its re-activations, where significantly shorter lifetimes less than 50 h were measured.

The storage lifetime could be increased for the p-GaN:Cs photocathode A5, which was thermally cleaned at 500 °C. For the p-GaN:Cs photocathode A6, cleaned at a temperature between 400 and 450 °C, the lifetime could also be increased. The storage lifetime was extrapolated to over 5000 h for the first cycle in our experiment.

In summary, the p-GaN:Cs photocathode A5, thermally cleaned at 500 °C, showed the highest QE (>10%), while the GaN:Cs photocathode A6, cleaned at 400–450 °C, showed the longest lifetime (>5000 h). The first activation cycle of sample A6 resulted in 9.5% QE and the longest lifetime, and thus treating the p-GaN thermally at 400–450 °C represented the best compromise to obtain the required QE and storage lifetime result.

#### 3.1.4. Activation Process

The p-GaN on sapphire sample A5 was wet chemically cleaned and then thermally cleaned at 500 °C. It was activated with a thin layer of cesium when it was cooled down to room temperature. The same thermal cleaning was repeated in every activation cycle.

In Figure 6a, typical photocurrent curves during the activation are shown. An aperture reduced the light power and consequently the photocurrent, which was a relief for the vacuum. In the beginning, no photocurrent could be detected after the cesium deposition had been started. After 10 to 20 min, the photocurrent started to rise slowly and steadily until it reached a maximum. At the end when the photocurrent reached a maximum, the aperture was opened to take the maximum photocurrent value at 110 µW light power. The starting points and speed of growth depend on the current applied to the cesium dispenser.

The QE decay curves after a successful activation for sample A5 are shown in Figure 6b. The QE dropped exponentially during the following hours after the activation and stabilized after a certain time. The same sample was thermally cleaned again when the QE reached 1%. After the thermal cleaning, before the sample was activated with cesium again, some photocurrent was observable. This indicates that the thermal cleaning does not remove all of the cesium from the p-GaN surface. We observed this phenomenon for samples A5 and A6, but not for sample A4, which was treated at a higher temperature.

The QE in the first cycle reached 11% for sample A5, which was reproducible in the second, third and fourth cycles. Thus, sample A5 showed the same QE value and QE decay independently of the number of activation cycles.

An extrapolated lifetime of over 2000 h in the third cycle was measured, which fulfills the requirement for using p-GaN:Cs in particle accelerator photoinjectors.

#### 3.1.5. Surface Morphology after Thermal Cleaning and Activation

The cleaned and activated p-GaN:Cs photocathodes were studied with SEM in order to examine their surface morphology. Additionally, the surface composition was characterized by EDX spectroscopy. In fact, the p-GaN:Cs samples were taken out from the UHV chamber. Therefore, the resulting EDX spectra might not show the original surface conditions because the p-GaN:Cs photocathodes were exposed to air. In particular, the cesium oxidizes into a cesium oxide or cesium hydroxide compound, and thus the original surface composition changes.

Figure 7a shows an SEM image of a part of the 9 × 9 mm² p-GaN:Cs photocathode (A1). Sample A1 underwent four cycles of thermal cleaning above 600 °C, each followed by a cesium activations. Therefore, sample A1 underwent the same treatment as sample A4. The different brightnesses in the SEM image indicate an inhomogeneous p-GaN:Cs surface. The half-round, bright area on the left side of the image derives from a round steel washer, which was used to fix the p-GaN sample on our sample holder. The original p-GaN:Cs surface on the right side in Figure 7a appears darker. Therefore, the brighter area on the left side contains heavier elements compared to the rest of the p-GaN:Cs surface. Iron was found in the bright area where the washer was located. This is confirmed by EDX measurements, shown in Appendix A.

The p-GaN:Cs photocathode A5 was thermally cleaned at 500 °C and then activated for four cycles. A part of the A5 surface is shown in the SEM image in Figure 7b. This image was taken additionally to compare the influence of the thermal cleaning on the surface morphology with sample A1, shown in Figure 7a. Therefore, both SEM images were taken in the area where the samples have been fixed with a steel washer. In Figure 7b, the surface of sample A5 appears homogeneous. In addition, other parts of sample A5 were examined with SEM, but no discrepancies could be observed. The surface composition was proven by an EDX measurement, showing an expected ratio of 1:1 for Ga:N, which is shown in Appendix A. Thus, the SEM/EDX measurements have reinforced the statement that the thermal cleaning at temperatures up to 500 °C neither changes the original p-GaN surface nor destroys its original composition.

### 3.2. p-GaN on Silicon

In our study, p-GaN on silicon was also examined for comparison with p-GaN on sapphire. Sapphire presents a challenge for future heat dissipation in photoinjectors as an insulator. Therefore, p-GaN on silicon samples were also considered for this study as silicon provides a better conductivity than sapphire.

For reasonable comparison, the p-GaN on silicon samples were treated in the same way as the p-GaN on sapphire samples, as explained above in Section 3.1. We started with AFM measurements before and after wet chemical cleaning, we continued with a thermal cleaning at various temperatures and cesium activation, and finally we applied SEM to study the surface morphology.

#### 3.2.1. Surface Topography of Native Samples

An overview AFM image of the native, untreated p-GaN on silicon surface is shown in Figure 8. The original p-GaN on silicon exhibits a smooth and uniform surface. Only a few particles and dust contaminations were found, resulting in an RMS roughness of about 3.0 nm.

#### 3.2.2. Wet Chemical Cleaning

The p-GaN on silicon samples were wet chemically cleaned with piranha solution, and the etched surface topography was examined by AFM, as shown in Figure 9.

The wet chemical cleaning results in the removal of contaminants and oxide layers on the sample surface. Consequently, the terrace-like structure of p-GaN on silicon could be observed. This terrace-like structure is shown in Figure 9, which derives from the different speeds of crystal growth on the AIN buffer layer. Therefore, the p-GaN on silicon shows a different structure than p-GaN on sapphire (Figure 3). The black spots in Figure 9b are hexagonal etching pits, exactly representing the hexagonal structure of the grown p-GaN. The p-GaN on silicon shows significantly more etching pits than p-GaN on sapphire, which indicates a higher defect number in the crystal lattice.

Owing to the wet chemical cleaning, the RMS roughness of p-GaN on silicon could be reduced to about 0.5 nm, which is an enormous improvement and beneficial for future applications in the injector system.

#### 3.2.3. Influence of Thermal Cleaning

After they have been wet chemically cleaned, the p-GaN on silicon samples were transferred under a nitrogen atmosphere into the UHV chamber, where they underwent a thermal cleaning process with the purpose of obtaining an atomically clean surface before cesium deposition.

The gained QE values for different p-GaN:Cs on silicon photocathodes, which had been thermally cleaned at various temperatures, are shown in Figure 10. The thermal cleaning followed by the activation was repeated several times for each sample.

Sample B1 (violet) showed the lowest QE and was destroyed by the thermal cleaning above 600 °C. Therefore, no re-activation could be carried out for B1. Sample B2 (red), cleaned at temperatures of 500 °C, demonstrated a slightly higher QE (2.5%) which was reproduced for five activation cycles.

However, the best QE (4%) was achieved with sample B3 (yellow), which was thermally cleaned at 400 °C and activated for three cycles.

Additionally, the storage lifetimes of the p-GaN:Cs on silicon samples were also examined and are shown in Figure 11. For sample B1, no lifetime could be given because the sample was destroyed after the thermal cleaning and the activation. We assume that the thermal cleaning at this high temperature caused crystal tension, so the sample burst after cesium was applied to the surface.

Samples B2 and B3 show much shorter lifetimes compared to those of p-GaN:Cs on sapphire, shown in Section 3.1.3 in Figure 5. This is assumed to be due to the high lattice mismatch between silicon and p-GaN that is not fully compensated by the AIN buffer layer.

During the lifetime measurement, the activated p-GaN:Cs on silicon photocathodes were stored in the UHV with an average pressure of 5 × 10^−10^ mbar.

The QE decay of samples B2 and B3 was followed for several hours and days. Sample B2, which was cleaned at 500 °C, showed lifetimes between 30 and 120 h. Its lifetime increased with every activation cycle. Therefore, the longest lifetime with almost 120 h was achieved in the fifth cycle. In contrast, the storage lifetime of sample B3, which was cleaned at 400 °C, was between 50 and 60 h and similar for each activation cycle.

In summary, sample B2, cleaned at 500 °C, showed a lower QE but a longer lifetime than sample B3, thermally cleaned at 400 °C. Sample B1, cleaned above 600 °C, showed the lowest QE and was destroyed at this temperature.

In accordance with the results obtained for p-GaN:Cs on sapphire photocathodes (see Section 3.1.3 above), a thermal cleaning at relatively lower temperatures is more beneficial for the QE and the lifetime of p-GaN:Cs on silicon photocathodes.

However, p-GaN:Cs photocathodes on silicon show significantly lower QE and lifetimes than p-GaN:Cs photocathodes on sapphire. This is assumed to be due to the high lattice mismatch between silicon and p-GaN that is not fully compensated by the AIN buffer layer.

#### 3.2.4. Activation Process

Photocurrent curves of p-GaN on silicon for sample B2, thermally cleaned at 500 °C, are shown in Figure 12a. The photocurrent curve progression is similar for each of the five activation cycles. The photocurrent starts to increase after about 15–25 min and reaches a maximum value, where the deposition of cesium is stopped.

All activation cycles were carried out at the same cesium flux, and thus the photocurrent curves are similar to each other. As mentioned already in Section 3.1.4, an aperture reduced the light power and consequently the photocurrent. After the cesium deposition was finished, the aperture was opened in order to receive the maximum photocurrent at 110 µW light power.

Figure 12b shows the corresponding QE decay curves for the five activation cycles of p-GaN:Cs on silicon photocathode B2. The QE values and the decays were reproducible and similar. As already mentioned, the lifetime increased with the increase in the number of activation cycles, and thus the fifth activation cycle showed the longest lifetime and the slowest QE decay.

#### 3.2.5. Surface Morphology after Thermal Cleaning and Cesium Activation

The surface morphology of the p-GaN:Cs on silicon photocathode (sample B2), which was thermally cleaned at 500 °C and activated five times, was examined by SEM, as shown in Figure 13.

The examined p-GaN:Cs photocathode (sample B2) was taken out of the UHV chamber and exposed to air. Thus, the shown SEM images and corresponding EDX results might not represent the original surface, as already mentioned in Section 3.1.5. Surface components, such as the cesium, were probably oxidized. 

In general, Figure 13 shows a smooth and uniform p-GaN:Cs surface. Although it is difficult to examine monolayer thin films with SEM and EDX, cesium oxide could be detected here. The cesium oxide appears as round and very small particles and could be found all over the surface. The cesium was oxidized to a cesium oxide or a cesium hydroxide due to the exposure to natural air and moisture. Evidence is given by EDX spectroscopy in Appendix A.

The p-GaN:Cs on silicon surface of sample B2 appears homogeneous, and no significant irregularities could be observed. The thermal cleaning at 500 °C is proven to be safe for the p-GaN surface and leads to the effective production of a p-GaN:Cs photocathode.

## 4. Conclusions

In our experiments, we succeeded in the cleaning and the activation of two types of commercially available p-GaN samples, grown on sapphire and on silicon, respectively. The presence of O_2_ during the sample handling, as well as the temperature applied during the thermal cleaning, influenced the surface morphology significantly and affected the QE and lifetime. The activation method exclusively with cesium is feasible and practically reliable. With p-GaN:Cs on sapphire photocathodes, a high QE of over 10% and an extrapolated storage lifetime of over 5000 h have been demonstrated.

The wet chemical cleaning process was validated via AFM measurements, showing that the cleaned surface possesses an average RMS roughness of 1.7 nm for p-GaN samples grown on sapphire and 0.5 nm for those grown on silicon substrates. For both types, p-GaN on sapphire and on silicon, etching pits that represent the hexagonal crystal structure were observed.

Furthermore, AFM measurements confirmed the adsorption of oxygen when the freshly cleaned samples were exposed to air. Consequently, it is crucial to protect the p-GaN sample from the air environment to avoid such oxidization. Thus, the cleaning process was carried out under dry nitrogen in a glovebox, and the p-GaN samples were further transported under a nitrogen atmosphere into a UHV chamber.

For p-GaN on silicon with an AIN buffer layer in between, the wet chemical cleaning leads to the successful removal of dust particles and reveals the typical terrace-like structure.

After the wet chemical cleaning, the p-GaN samples underwent a thermal cleaning in a UHV chamber. Treatments at different temperatures resulted in various QE values and storage lifetimes. Moderate temperatures of 400–500 °C were found to be more beneficial for the p-GaN surface quality, which was reflected by higher QE values achieved for both substrate materials. SEM investigations showed that the surface morphology of p-GaN was destroyed at higher temperatures.

After the thermal cleaning, the samples were activated with a thin layer of cesium at an average pressure of 1 × 10^−9^ mbar. The highest QE could be achieved for p-GaN:Cs on sapphire at 11% and for p-GaN:Cs on silicon at 4%. The storage lifetimes for p-GaN:Cs on silicon were significantly shorter than those achieved for p-GaN:Cs on sapphire, which is related to the p-GaN crystal quality, influenced by the substrate.

Furthermore, our study showed that the re-activation of p-GaN:Cs photocathodes is practical and reproducible at least five times.

The surface morphology was studied with SEM and EDX after the samples were thermally cleaned and activated with cesium. The results showed that the surface appeared inhomogeneous when the sample was cleaned at a high temperature above 600 °C. It is assumed that the thermal cleaning from the front side via a halogen lamp might not be the best choice for obtaining an atomically clean surface. A thermal cleaning from the back side through the substrate would be another possibility with the aim of protecting the p-GaN surface.

For the re-activations, we measured some photocurrent for the p-GaN on sapphire samples after they were thermally cleaned again, prior to the next cesium activation. This phenomenon was observed only for samples treated at 500 °C and 400 °C and not for the samples that were thermally treated at higher temperatures. This phenomenon indicates that cesium was not removed completely from the surface. This could be detectable with the help of a surface analysis tool without leaving a UHV regime. We plan to connect an X-ray photoelectron spectrometer (XPS) to our preparation system in order to follow and understand the changes in the surface electronic states before, during and after any treatment of p-GaN:Cs photocathodes. These in situ XPS results will be reported in forthcoming publications.

## Figures and Tables

**Figure 1 micromachines-13-00849-f001:**
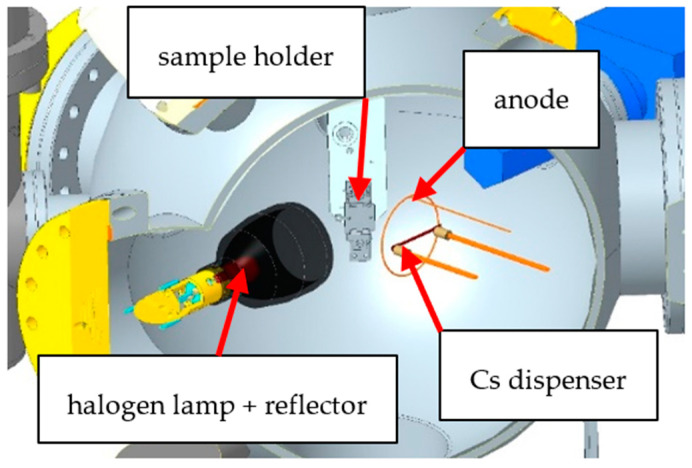
Scheme of the UHV chamber, showing the sample holder, the halogen lamp with reflector, the copper anode and the cesium (Cs) dispenser.

**Figure 2 micromachines-13-00849-f002:**
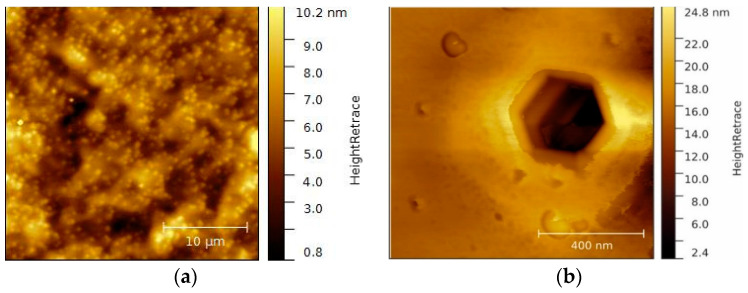
(**a**) A 25 × 25 µm² overview AFM image of native p-GaN on sapphire and (**b**) a higher magnification 1 × 1 µm² AFM image of an etching pit on the same surface.

**Figure 3 micromachines-13-00849-f003:**
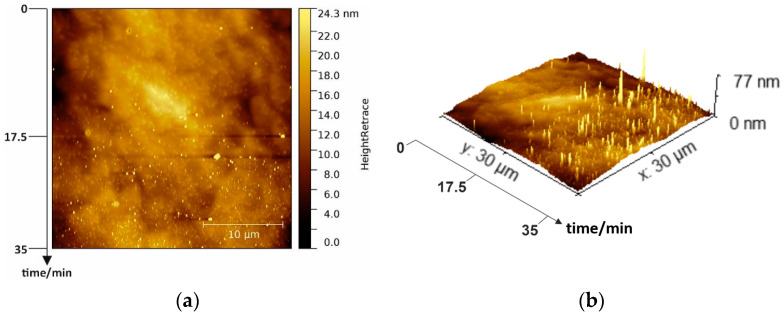
(**a**) A 30 × 30 µm² overview 2D AFM image of wet chemically cleaned p-GaN and (**b**) corresponding 3D image.

**Figure 4 micromachines-13-00849-f004:**
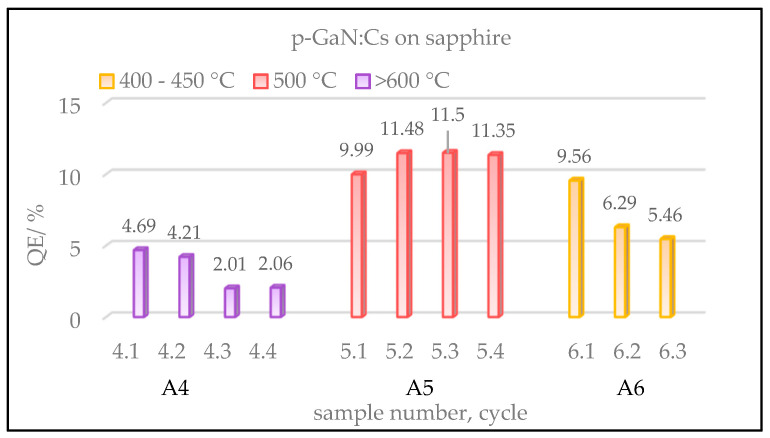
Gained QE values of p-GaN:Cs on sapphire photocathodes, depending on various temperatures used in the thermal cleaning. Sample A4 (violet) was cleaned at over 600 °C and activated for four cycles. Sample A5 (red) was cleaned at 500 °C and activated for four cycles, while sample A6 (yellow) was cleaned at 400–450 °C and activated for three cycles.

**Figure 5 micromachines-13-00849-f005:**
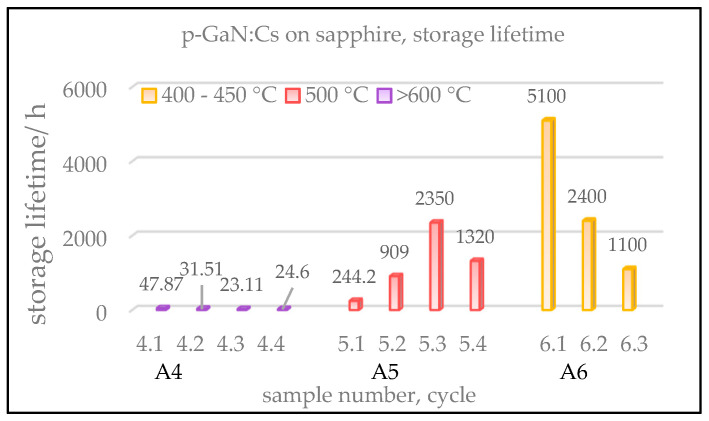
Extrapolated storage lifetimes of p-GaN:Cs on sapphire photocathodes, influenced by the cleaning temperature. The numeration of samples is the same as in Figure 4. Equation (2) was used for the extrapolation of the lifetimes.

**Figure 6 micromachines-13-00849-f006:**
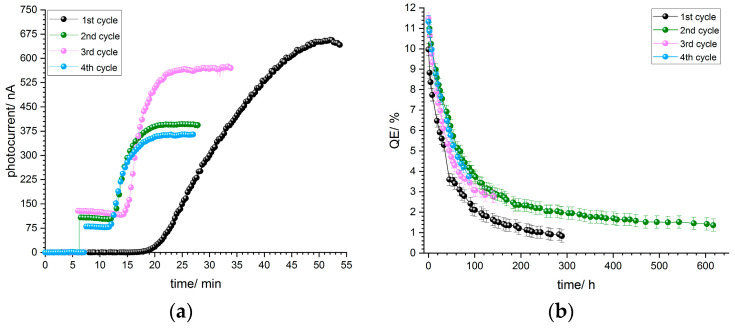
(**a**) Photocurrent curves of p-GaN (sample A5), which was thermally cleaned at 500 °C and activated with cesium for four cycles, and (**b**) QE decay curves of the same p-GaN:Cs photocathode after activation.

**Figure 7 micromachines-13-00849-f007:**
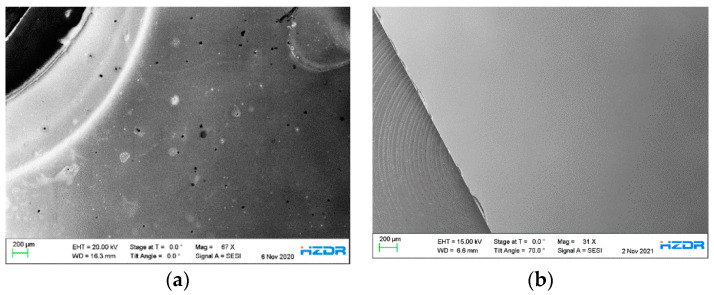
(**a**) SEM overview image of a part of the p-GaN:Cs photocathode (sample A1), which was thermally cleaned above 600 °C and activated for four cycles; (**b**) SEM overview image of a part of the p-GaN:Cs photocathode (sample A5), which was thermally cleaned at 500 °C and activated for five cycles.

**Figure 8 micromachines-13-00849-f008:**
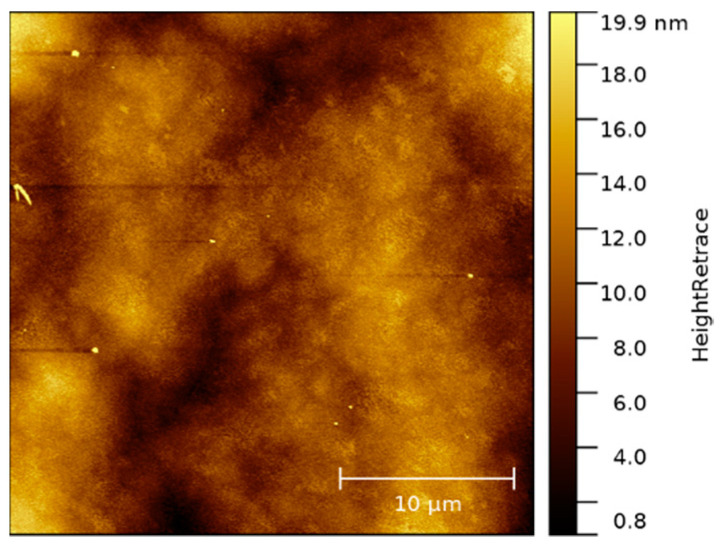
A 25 × 25 µm² overview AFM image of the native surface of as-received p-GaN on silicon.

**Figure 9 micromachines-13-00849-f009:**
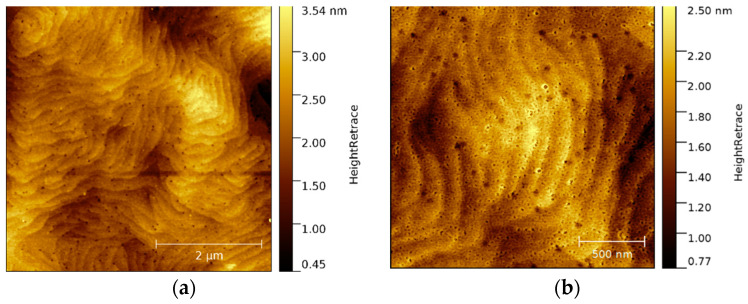
(**a**) A 5 × 5 µm² AFM image of the wet chemically cleaned p-GaN on silicon surface and (**b**) a higher magnification 1 × 1 nm² AFM image of the same sample.

**Figure 10 micromachines-13-00849-f010:**
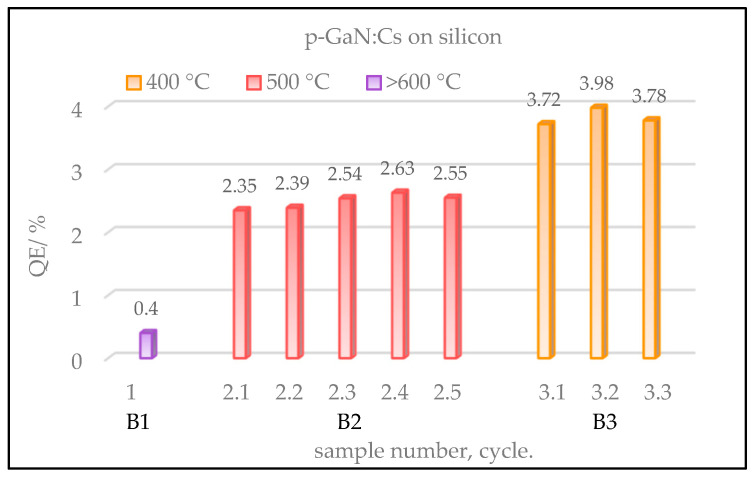
Gained QE values of p-GaN:Cs on silicon photocathodes, depending on various temperatures used in the thermal cleaning. Sample B1 (violet) was thermally cleaned above 600 °C and activated only once. Sample B3 (yellow), which was cleaned at 400 °C, achieved the best QE in three cycles. At the same time, sample B2 (red), which was thermally cleaned at 500 °C, showed relatively lower QE than B3, but a higher reproducibility in five cycles.

**Figure 11 micromachines-13-00849-f011:**
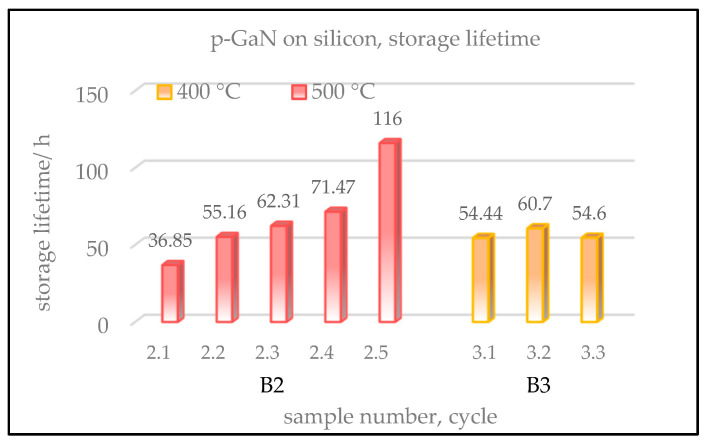
Storage lifetimes of p-GaN:Cs on silicon photocathodes. Sample descriptions are the same as in Figure 10.

**Figure 12 micromachines-13-00849-f012:**
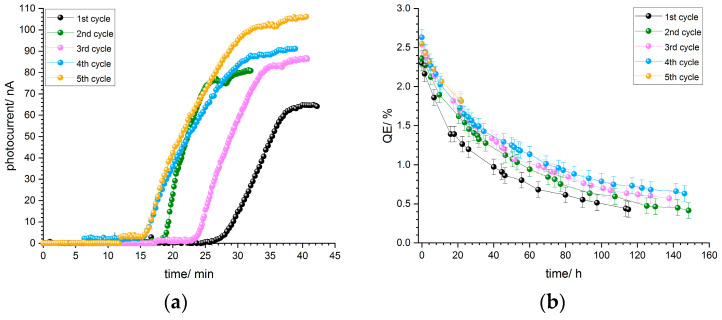
(**a**) Photocurrent curves of the five activations for p-GaN:Cs on silicon photocathode B2 and (**b**) QE decay curves for each activation cycle of the same sample.

**Figure 13 micromachines-13-00849-f013:**
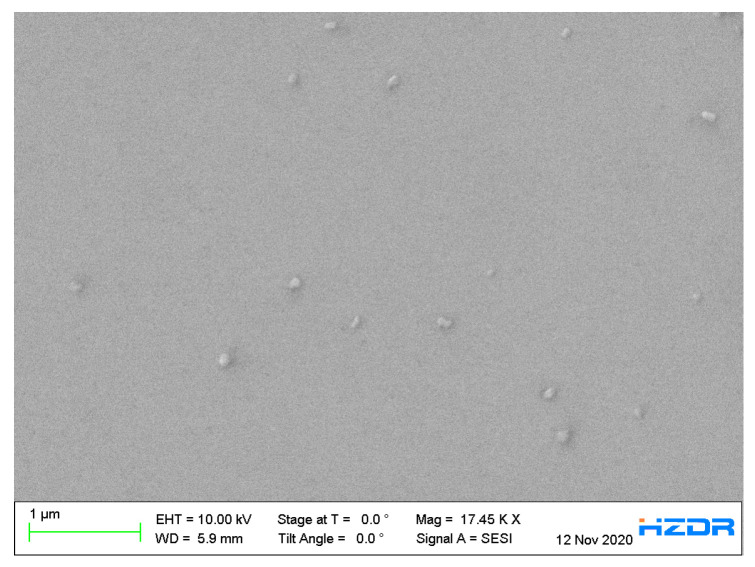
SEM image of p-GaN:Cs on silicon photocathode (sample B2), thermally cleaned at 500 °C and activated with cesium for five cycles.

## Data Availability

Not applicable.

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
