# Peer review of "Influence of Surface Cleaning on Quantum Efficiency, Lifetime and Surface Morphology of p-GaN:Cs Photocathodes"

_micromachines, 2022, doi:10.3390/mi13060849_

Round 1
Reviewer 1 Report
This paper did a detailed research on the p-GaN photocathode on sapphire and silicon, which is very interested for the researchers in this area. This new kind of photocathode will help to fulfill the demand of XFEL and other applications such as high brightness electron sources. For this paper, discussion is needed for the different performance between p-GaN photocathode on sapphire and silicon. And more attention should be paid to the lifetim. The English of the paper should be checked again.

Author Response
Dear Reviewer,
thank you very much for reading this article and your fruitful suggestions. I would like to answer your questions (you can also see in the attachment):
- Line 17: „in respect to“ is corrected to „with respect to“
- Line 38: „were“ is corrected into „are“
- Line 41: K2CsSb is changed into K2SbCs
- Line 66: step is changed into steps
- Line 77: space added
- Line 95: The IR sensor was calibrated by the supplier, especially for semiconductor surfaces. I added the company and the model type in line 96
- Line 109: I added the evaporation current in line 111 (it was 3.3 A – 3.5 A)
- Line 110: I added the company and model type for the UV-LED
- Line 114: we used only cesium in this activation processes. We also tried Yoyo activation, but the results were not better than with only cesium because we did not use cesium and oxygen excessively.
The sample was activated at room temperature. We have no data for an activation with an hot substrate - Line 151: This etching pit is an indicator for a defect in the crystal lattice, it means that a surface with many etching pits has a poor crystallinity. We do not have the possibility to do a QE map at the moment, but it is planned for future.
- Line 194: space added
- Line 206: „can been seen“ is corrected to „can be seen“
- Line 230-237: inset of the temperature is corrected to >600 °C
- Line 242: We tried to keep one temperature for one sample to guarantee a higher statistic accuracy and repeatable data. We also tried different temperatures on one sample and publish these results in the forthcoming publications together with XPS results
- Line 246: QE < 10 % is corrected into QE > 10 %
- Line 271: I added the reference [32] to the equation (2)
- Line 276: the fluctuations in the vacuum are considered normal, influenced by the normal environmental fluctuation in the lab
- Line 290: 9,5 % is corrected into 9.5%
- Line 300: Yoyo process was also tried, but the QE and lifetime were not better than an the results with with only cesium. It will be published additionally.
- Line 310: Yes indeed. An aperture reduced the photocurrent in order to relieve our vacuum. We recognized that the vacuum increases when the photocathode is illuminated with the full light power of 110 µW. Therefore, we limit the power by an aperture and open it fully when the activation was finished to receive the maximum photocurrent value. I added a few sentences in the text.
- Line 303: „raise“ is changed into „rise“
- Line 329: We saw this phenomenon for samples when they were treated at lower temperature and now with XPS we have evidence that not all cesium is removed. When the samples are treated at higher temperature, such as 600 °C, we did not observe any photocurrent in the re-activation. Therefore, the cesium is removed completely before the sample was re-activated again.
- Line 320: to Fig. 6b: yes we tried to save time. We did not measure the lifetime for all cycles till the QE was back to 1%. We decided to measure it in shorter time and to do an extrapolation of the storage lifetime as described in line 270.
- Lines 345-356: Fig. 7a shows indeed sample A1. Sample A1 was treated in the same way as A4. Therefore, we assume the surface morphology is the same for both. A1 showed same QE and lifetimes as A4, but we decided not to show A1-A3 in this publication. Three samples for sapphire and three for silicon. In fact, I measured more samples. I took A1 to the SEM because it was already ejected from the UHV chamber and I got appointment at SEM. It takes a long time to receive an appointment because we do not have an own SEM. This is also the reason why there was some time in between the measurements. We decided not to measure A6, treated at 400 °C, because A5 did not show any irregularities on the surface. We expect that A6 show also no irregularities. Now with XPS, we do have a better tool to do measure the surface composition without leaving UHV.
- Line 370: I corrected the heating cycles to „four“
- Line 367: see comment 24)
- Line 359: Sample A1 is indeed sample A1
- Line 345: Fig. 7b: This is exactly the point. You cannot see the effect of the washer on A5. The temperature was suitable so that no iron was transferred and the p-GaN composition was not destroyed. We proofed all edges and did additionally EDX. Refer to line 377.
- Line 345-356: Yes indeed. We explained in lines 370-381 that in fact the surface composition and the homogeneity are beneficial for higher QE. Compared to A1 were the composition changed and the morphology shows irregularities, which resulted in less QE.
- Line 389: p-GaN on silicon was treated in all steps in the same way as p-GaN on sapphire, including wet chemical treatment, etc. Refer to line 389.
- Line 427: I changed Figure 10 into Figure 9
- Line 410: yes I observed the effect of etched samples in the atmosphere, but it was not possible to observe this effect for p-GaN on silicon like in Fig. 3.
- Line 438: yes the activation process is the same, for both materials. Refer to line 389.
- Line 442: space added
- Line 451-461: Fig. 10: I changed <600 °C into > 600°C
- Line 470: Yes indeed. The main reason for the busted GaN in silcon was the thermal heating. We assume that the surface cracked and after applying cesium it busted because of surface tension. I added an short explanation in line 473.
- Line 490: the vacuum has a normal fluctuation due to a natural fluctuation of the environmental temperature. These little fluctuation in -10 mbar range should have no influence on the photocathodes‘ performance.
- Line 448: space added
- Line 524: Fig. 12: There is no QE left after the thermal heating for GaN on silicon. This means that the cesium was desorbed completely from its surface. A reason might be that the cesium was less weakly adsorbed on p-GaN on silcon than on p-GaN on sapphire. It could also be that the p-GaN on silicon surface had some contaminations on its surface, which distrubed the cesium. We can check this with XPS measurements in future.
- Line 578: space added
- Line 612: I rewrote the sentence.

Author Response
Dear Reviewer,
thank you very much for reading the article and for your fruitful suggestions. I would like to answer your questions (see also in the attachment):
- Line 192 to 196: Unfortunately we are not able to provide AFM images under nitrogen atmosphere. With our microscope we can only measure in normal atmosphere or in liquids. In the future we are able to proof such oxidization with XPS measurements.
- Figures 4,5 and 10 and 11: I changed the inset of the temperature to > 600°C
- Line 385: I added a few sentences for the necessarity of p-GaN on silcon. The main reason why we decided to look for other substrates is that sapphire is an insulator and cannot be used for high charging applications. This might cause heat dissipation problems in the photoinjector.
- Line 77: I added some descriptions that the low QE and lifetime seems influenced by the poor crystallinity of p-GaN on silicon in line 430, 473, 503 and in the conclusion.

Round 2
Reviewer 2 Report
accepted